# Enhancing Reasoning Chains through Quasi-GANs and Textual Gradient Feedback

## Abstract

Large language models (LLMs) have recently advanced reasoning in multi-agent systems (MAS), yet existing work mainly focuses on improving forward reasoning accuracy, overlooking the potential of adversarial mechanisms with backward generation of erroneous reasoning chains to enhance both accuracy and stability. We propose a novel adversarial learning framework in which a forward generator produces accurate reasoning chains, while a backward generator constructs adversarial erroneous chains. Guided by a discriminator providing gradient feedback in the textual domain, both generators iteratively refine their outputs through competitive optimization with generative adversarial networks (GANs). This competitive optimization reduces variability in outputs for identical queries, increases robustness to prompt perturbations, and provides interpretability into the distinct roles of the two generators by dynamically tracking the evolution of reasoning chains. Experiments show that, after two to three rounds of prompt optimization, our method improves reasoning accuracy from 73.7% to 81.6%, and reduces instability from 0.39 to 0.08. These results demonstrate the proposed framework's ability to jointly optimize accuracy and stability, and highlight the promise of adversarial forward-backward mechanisms in advancing multi-agent reasoning systems.

## 1 Introduction

In recent years, large language models (LLMs) have demonstrated remarkable capabilities in reasoning tasks, particularly through efficient reasoning optimization methods such as Chain-of-Thought (CoT) prompting (Kojima et al., 2022). Leveraging these advances, LLM-powered AI agents have exhibited strong abilities in tool usage, self-reflection, and other reasoning-related functionalities. Multi-agent systems (MAS) further amplify these capabilities by enabling collaboration among multiple AI agents, significantly improving both the efficiency and accuracy of reasoning tasks.

However, as task complexity increases, LLM-based systems often experience performance degradation, while LLM-powered MAS (LLM-MAS) exhibit notable instability in reasoning. Specifically, for the same prompt on the same problem, the system may produce a correct reasoning chain in one run and an incorrect one in another. Such variability undermines the overall accuracy and reliability of the reasoning process. Most existing work focuses on improving the accuracy of forward reasoning chains by guiding the generation of intermediate reasoning steps. Methods include prompt optimization engineering, model fine-tuning, supervised intermediate steps, or sampling multiple reasoning trajectories. Nevertheless, these techniques remain sensitive to prompt-level noise. Even small changes in the prompt can lead to divergent intermediate conclusions and even reversed final answers. This indicates that simply extending forward reasoning chains is insufficient to achieve the global consistency required for stable and accurate reasoning.

To address this limitation, we propose an innovative adversarial generation mechanism for LLM reasoning. Our framework introduces generative adversarial network (GAN)-based methods, where a backward generator deliberately produces erroneous reasoning chains, containing logical leaps, conceptual confusions, or false assumptions, to challenge the forward generator. A discriminator evaluates the quality of both forward and backward reasoning chains, providing targeted feedback that enables generators to adjust their reasoning strategies accordingly. To further enhance training effectiveness, we integrate a textual gradient technique into the discriminator, replacing conventional numerical gradient updates with natural language feedback.

Extensive experiments have verified that our method performs excellently across multiple datasets for common and adversarial-like non-fine-tuning reasoning enhancement methods, achieving an average accuracy of 81.6%, demonstrating stability after fewer than five iterations, while also improving explainability as the evolution of each reasoning chain becomes more transparent and traceable.

Our main contributions are summarized as follows:

- Introducing a backward generator that produces misleading reasoning chains to challenge the forward generator substantially reduces instability in complex MAS reasoning tasks and improves accuracy and consistency.

- Designing a transparent adversarial reasoning system enables joint optimization of forward and backward generators through adversarial training. This framework improves reasoning-chain quality, stability, and adaptability, while its transparency enables step-level traceability and optimization via textual feedback, thereby enhancing interpretability.

## 2 RELATED WORK

### 2.1 REASONING IN LLM-BASED MULTI-AGENT SYSTEMS

Recent advancements in LLMs have motivated the development of multi-agent systems (MAS) for complex reasoning tasks. Existing studies primarily focus on optimizing the accuracy of forward-generated reasoning chains, employing various techniques such as Chain-of-Thought (Kojima et al., 2022) and Self-Refine (Madaan et al., 2023) to enhance the reasoning capabilities of models. In MAS, multiple agents collaborate through communication and role specialization, leading to more reliable reasoning processes compared to single-agent setups. Surveys such as Guo et al. (2024); Tran et al. (2025) have outlined the landscape of LLM-based MAS, highlighting their applications in task solving, simulation, and evaluation.

A key direction is to design structured reasoning processes among agents. For example, Motwani et al. (2025) introduced the MALT framework, where heterogeneous roles (generator, verifier, refiner) are organized into a reasoning search tree and optimized through trajectory-level updates. From a game-theoretic perspective, Yi et al. (2025b) proposed ECON, which models rational multi-agent decision-making as Bayesian Nash equilibria, thereby improving cooperative reasoning efficiency. Meanwhile, robustness and safety concerns have also gained attention. Ebrahimi et al. (2025) proposed credibility scoring to mitigate the impact of malicious or low-quality agents, further enhancing system-level reasoning.

### 2.2 ADVERSARIAL TRAINING MECHANISMS

Adversarial training has long been employed to improve model robustness and generalization. Classical works such as GANs (Goodfellow et al., 2020) have inspired extensions across modalities, including StyleGAN3 (Karras et al., 2021) with frequency-domain regularization, PAIRED (Dennis et al., 2020) for environment-based opponent modeling, and AdvGAN (Xiao et al., 2018) for sample-level adversarial attacks.

In the context of LLM reasoning, adversarial mechanisms have recently been applied to LLM reasoning, where adversarial agents generate misleading reasoning chains or counterexamples to expose weaknesses, enhancing robustness across diverse application domains. For example, red-teaming approaches (Perez et al., 2022; Ganguli et al., 2022) and adversarial prompting (Zhao et al., 2024b) improve robustness in LLM reasoning, while debate-style adversarial collaboration improves factuality and reduces hallucination (Yang et al., 2025b). For MAS, reasoning tasks often involve collaboration and information sharing among multiple agents, where traditional reasoning enhancement methods face certain limitations. By introducing adversarial mechanisms, a backward generator can produce misleading reasoning chains to challenge the forward reasoning process, thereby helping the system identify potential errors and ultimately enhancing its stability and adaptability.

## 3 METHOD

### 3.1 REASONING CHAIN ENHANCEMENT BASED ON QUASI-GAN METHODOLOGY

We propose a reasoning chain enhancement framework based on Quasi-GAN methodology, consisting of a forward generator $G_1$ and a backward generator $G_2$. The core idea is to improve the robustness of $G_1$ by challenging it with adversarial reasoning chains produced by $G_2$.

Specifically, $G_1$ generates forward reasoning chains aimed at solving the task, while $G_2$ deliberately produces erroneous chains with logical jumps or false assumptions designed to expose weaknesses in the reasoning process of $G_1$. During this adversarial process, a discriminator $D$ mediates the interaction of the dual-generator by evaluating the quality of both chains and providing natural-language feedback rather than numeric gradients. We adopt a textual gradient optimization scheme, where this feedback is used to indirectly update both $G_1$ and $G_2$, encouraging $G_1$ to refine its reasoning under adversarial pressure and $G_2$ to generate more effective challenges. Additionally, $D$ maintains a record of the latest optimization outcomes to continuously refine its own evaluation strategy, ensuring effective multi-round adversarial training. This adversarial architecture allows the forward generator $G_1$ to iteratively improve accuracy and stability, the backward generator $G_2$ to adaptively craft harder counterexamples, and the discriminator $D$ to evolve as a more reliable feedback provider over time.

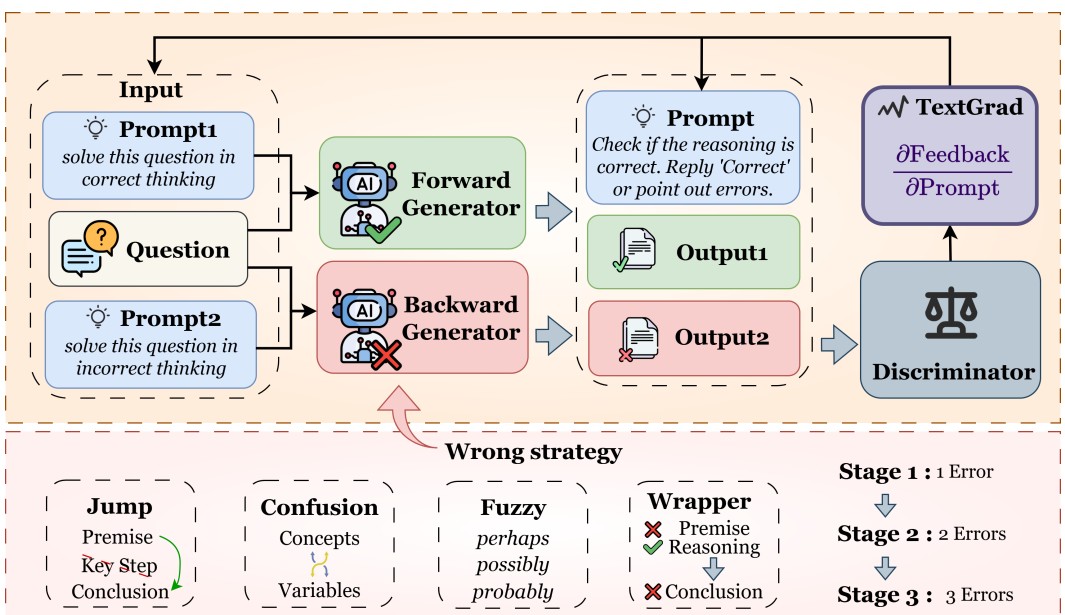

Figure 1: Overview of the proposed framework. The framework architecture featuring dual generators (correct and error-inducing), a discriminator evaluating reasoning quality, and gradient-based prompt optimization. The system identifies and corrects four common reasoning error types through progressive training stages.

### 3.1.1 DUAL-GENERATOR AND DISCRIMINATOR IN QUASI-GAN METHODOLOGY

**Forward generator $G_1$:**

The task of $G_1$ is to produce a reasonable, coherent, and logically consistent Chain-of-Thought (CoT) reasoning sequence given an input question. Let the input question be denoted as $Q$, a general prompt assisting $G_1$ to produce correct answer be $P_1$, and the reasoning chain generated by $G_1$ be $C_1$, which can be formulated as:

$$C_1 = G_1(P_1 + Q, \theta_1), \tag{1}$$

where $\theta_1$ represents the parameters of $G_1$. And $G_1$ is prompted to generate logically consistent and interpretable reasoning steps, thereby ensuring the quality of the reasoning chain.

**Backward generator** $G_2$:

$G_2$ aims to produce an incorrect reasoning chain that stands in contrast to the reasoning chain generated by $G_1$, challenging $G_1$ with reasoning chains that contain misleading or flawed reasoning steps. Similar to $G_1$, let the output of $G_2$ of the same question $Q$ and a reversed prompt $P_2$ be denoted as $C_2$, which can be expressed as:

$$C_2 = G_2(P_2 + Q, \theta_2),\tag{2}$$

where $\theta_2$ corresponds to the parameters of $G_2$. The reasoning chains $C_2$ may involve logical leaps, conceptual confusions, or unsupported assumptions, intentionally designed to undermine the reasoning capability of $G_1$.

**Discriminator** $D$:

$D$ is responsible for evaluating the reasoning chains generated by $G_1$ and $G_2$ and providing textual gradient feedback. By analyzing the logical soundness of each reasoning chain, $D$ generates feedback aimed at guiding $G_1$ to refine its reasoning process. The outputs of the discriminator include quality assessments for reasoning chains from both $G_1$ and $G_2$, formally expressed as:

$$F_1 = D(P_1 + Q, C_1, \theta_D),\tag{3}$$

$$F_2 = D(P_2 + Q, C_2, \theta_D),\tag{4}$$

where $\theta_D$ denotes the parameters of the discriminator, and $F_1$ and $F_2$ represent the textual gradient feedback for the forward reasoning chain $C_1$ and the adversarial reasoning chain $C_2$, respectively.

### 3.2 Adversarial Interference Strategy for Backward Generator

#### 3.2.1 Erroneous reasoning chain types

To effectively challenge the reasoning capability of the forward generator and encourage it to produce more rigorous reasoning chains, we define four typical reasoning error types. These error types capture common deficiencies observed in reasoning processes, serving as a structured basis for improving the generator's robustness and accuracy when facing diverse reasoning tasks. The specific error types are defined as follows:

(1) **Jump error**: Omitting critical intermediate steps and directly reaching the conclusion, encouraging the generator to detect missing necessary derivations in the reasoning chain. (2) **Confusion error**: Mixing up concepts or variables like velocity vs. acceleration, guiding the generator to accurately distinguish between different concepts during reasoning. (3) **Fuzzy error**: Using uncertainty expressions such as "might" or "probably", challenging the generator to maintain clarity and determinism in the reasoning process. (4) **Wrapper error**: Reasoning appears plausible but is based on false premises, leading the generator to identify potential flaws in underlying assumptions and reinforce the logical soundness of the reasoning. By defining these four representative error types, we not only provide clear optimization targets for the generator but also ensure progressive improvement in reasoning quality under diverse challenges.

#### 3.2.2 Erroneous chain type generation scheduling

We design a stage-wise error-type scheduling strategy that gradually increases the complexity and diversity of error types, thereby improving robustness and enabling the generator to handle reasoning challenges of varying difficulty. The scheduling consists of three stages:

- **Stage 1** (Iterations 1 to 2): Single error type per iteration.
  At this stage, the generator focuses on addressing basic reasoning errors such as jump error and confusion error. Starting with simple errors allows the model to build fundamental detection and correction skills.

- **Stage 2** (Iterations 3 to 5): Combination of two error types per iteration.
  This stage increases training complexity and diversity, forcing the generator to maintain consistency and accuracy under multiple simultaneous disturbances.

- **Stage 3** (Iteration 6 onward): Combination of three error types per iteration.

At this stage, the complexity of adversarial reasoning chain generation is further intensified, challenging the generator to maintain robust performance against more sophisticated reasoning disruptions.

Through this stage-wise scheduling strategy, the forward generator is incrementally guided toward higher-quality reasoning. Each stage focuses on addressing specific error types or combinations, allowing the model to develop resilience to increasingly complex reasoning challenges. This not only improves training efficiency but also prevents excessive difficulty in the early phase, ensuring steady and reliable progress.

### 3.3 DISCRIMINATOR WITH TEXTUAL GRADIENT OPTIMIZATION

To optimize the dual LLM generators in our framework, we introduce a textual backpropagation mechanism to the discriminator, implemented through the TEXTGRAD framework (Hou et al., 2023). Inspired by the principle of automatic differentiation, textual gradient optimization combines traditional gradient-based optimization to optimize variables or parameters within a system using natural language feedback, particularly in the context of LLMs. Specifically, this method transforms the AI system into a computational graph, where each node corresponds to a system variable, such as code snippets, molecular structures, or reasoning steps. The values of these nodes are optimized through textual gradients expressed in natural language feedback. The textual gradient backpropagation particularly consists of three stages: computational graph representation, gradient calculation, and textual gradient descent.

**Computational graph representation**:

We first represent the LLM reasoning system with GAN as a computation graph, where operations like LLM invocation and numerical solving are treated as nodes in general. For our proposed LLM reasoning enhancement framework, we model the reasoning processes of the LLM-based generators $G_1$, $G_2$, and the discriminator $D$ into a computational graph, upon which gradient computation is performed across variables.

**Gradient calculation**:

The textual gradient optimization simulates backpropagation by interpreting natural language feedback as a form of gradient signal. For the feedback optimization of the discriminator for the forward generator, as shown in Eq. 1 and Eq. 3, we can compute the gradient of the feedback $F_1$ with respect to the forward input prompt $P_1$ through the generation of forward chain $C_1$ as follows:

$$\frac{\partial F_1}{\partial C_1} = \nabla_D(C_1, F_1), \tag{5}$$

which represents the gradient of the feedback of forward generator with respect to $G_1$ generated reasoning chain $C_1$. Then the gradient of $F_1$ to $P_1$ can be computed via the chain rule:

$$\frac{\partial F_1}{\partial P_1} = \frac{\partial F_1}{\partial C_1} \circ \frac{\partial C_1}{\partial P_1} = \nabla_G(P_1, C_1, \frac{\partial F_1}{\partial C_1}). \tag{6}$$

Similarly, based on Eq. 2 and Eq. 4, the gradient of feedback for backward generator $F_2$ to the backward prompt $P_2$ can be derived through the generation of adversarial chain $C_2$:

$$\frac{\partial F_2}{\partial P_2} = \frac{\partial F_2}{\partial C_2} \circ \frac{\partial C_2}{\partial P_2} = \nabla_G(P_2, C_2, \frac{\partial F_2}{\partial C_2}). \tag{7}$$

**Textual gradient descent**:

To optimize the prompt for better reasoning chain generation, we apply the Textual Gradient Descent (TGD) algorithm:

$$P \leftarrow \text{TGD.step}(P, \frac{\partial F}{\partial P}). \tag{8}$$

Here, the update direction is determined by natural language feedback from the LLM, which acts as a surrogate gradient to progressively refine the system's objective.

### 3.4 Adversarial Training and Optimization Objectives

In our framework, the forward generator $G_1$ and the backward generator $G_2$ are optimized through an adversarial mechanism, with the discriminator $D$ serving as a judge that provides textual feedback. The discriminator evaluates both the correct reasoning chain $C_1$ and the adversarial chain $C_2$ to guide the generators in improving reasoning quality. Specifically, $D$ provides feedback based on the interference effect of $C_2$, encouraging $G_1$ to refine $C_1$. For example, feedback on $C_1$ may include suggestions such as "Add more reasoning steps in Step 3" or "Include missing premises in Step 2", while feedback on $C_2$ may focus on how deliberately introduced errors challenge $C_1$, like "Introduce a false assumption in Step 2" or "Omit a critical reasoning step in Step 3". Both $G_1$ and $G_2$ update their generation strategies according to the feedback, adjusting their prompts to iteratively improve the quality of reasoning chains and increase robustness against interference.

#### 3.4.1 Optimization Objectives of Generators

We adopt a prompt optimization scheme, updating the generators' prompts using textual feedback from the discriminator. The forward generator $G_1$ aims to produce a valid reasoning chain $C_1$ that receives positive feedback from the discriminator. The backward generator $G_2$, in contrast, is designed to generate misleading reasoning chains $C_2$ that maximize negative feedback from the discriminator, thereby challenging $G_1$. Since the discriminator $D$ can distinguish between $C_1$ and $C_2$ based on $P_1$ and $P_2$, it can correctly identify valid from erroneous chains, thus avoiding harmful feedback caused by confusion between correct and incorrect reasoning. The objectives can be abstractly formulated as:

$$\mathcal{L}_1 = \mathbb{E}_{Q \sim P_1} \left[ D(Q, G_1(P_1 + Q, \theta_1)) \right], \tag{9}$$

$$\mathcal{L}_2 = \mathbb{E}_{Q \sim P_2} \left[ D(Q, G_2(P_2 + Q, \theta_2)) \right]. \tag{10}$$

Here, $\theta_1$ and $\theta_2$ denote the parameters of $G_1$ and $G_2$, respectively, where the most influential factor for our method is the temperature parameter of the LLM. In particular, the temperature directly controls the trade-off between determinism and diversity in the generated reasoning chains. A lower temperature encourages more stable and deterministic outputs, while a higher temperature promotes diversity and introduces more challenging or adversarial reasoning paths.

#### 3.4.2 Discriminator Self-Optimization and Feedback

To enhance the discriminator's evaluation capability, we design a self-optimization mechanism. The discriminator updates its prompt based on previous feedback records:

$$P_{new} = P_{old} + \nabla_P \mathcal{L}_D(F_D), \tag{11}$$

where $P_{new}$ and $P_{old}$ denote the updated and current prompts for $D$, and $\nabla_P$ represents the update direction derived from discriminator feedback $F_D$. $F_D$ not only guides the correct chain $C_1$ but also uses the adversarial chain $C_2$ to interfere with the forward generator. The feedback is structured in natural language as:

- For $C_1$, feedback improves the rigor and coherence of reasoning (e.g., "Add more reasoning steps in Step 3");

- For $C_2$, feedback intentionally introduces misleading elements to challenge $G_1$ (e.g., "Insert a false assumption in Step 2").

The dual generators update their prompts according to the feedback:

$$P_{new} = P_{old} + \nabla_P \left[ F_D(C_1, C_2) \right], \tag{12}$$

where $F_D(C_1, C_2)$ represents the textual feedback from $D$, including corrective guidance for the correct chain and adversarial suggestions from the erroneous chain.

Through this contrastive feedback mechanism, the generators iteratively refine their prompts in each round, thereby improving the quality of reasoning and enhancing robustness against complex adversarial perturbations.

## 4 EXPERIMENTS

This section evaluates our GAN-based multi-agent reasoning framework's effectiveness and generalization. We compare our framework's QA performance against existing LLM reasoning methods across various training settings, LLM backbones, and datasets. These analyses demonstrate the overall effectiveness of our method and the contribution of each component, with additional experimental analyses presented in the appendix.

### 4.1 EXPERIMENTAL SETUP

#### 4.1.1 DATASETS AND EVALUATION

We conduct experiments on six question-answering (QA) datasets that cover a diverse range of reasoning tasks. The MATH dataset (Hendrycks et al., 2021) tests advanced mathematical problem solving requiring complex symbolic reasoning, while GSM8K (Cobbe et al., 2021) features grade-school level math problems that evaluate model capability in basic mathematical reasoning and logical deduction. For assessing logical and algorithmic reasoning, we employ the Big Bench Hard (BBH) dataset (Suzgun et al., 2023). We also incorporate MMLU-CF (Zhao et al., 2024a), a diagnostics subset of MMLU focusing on commonsense and factual knowledge, alongside HotpotQA (Yang et al., 2018), which demands multi-hop question answering where models must integrate information across supporting evidence. Finally, LongBench (Bai et al., 2024) evaluates long-context reasoning with multiple-choice questions and context lengths ranging from 8k to 2M words, spanning diverse task categories including single-doc QA, multi-doc QA, long in-context learning, dialogue understanding, codebase comprehension, and structured data reasoning.

Collectively, these datasets form a thorough evaluation suite across mathematical reasoning, logic, factual knowledge, multi-hop inference, and long-context comprehension, validating our framework's effectiveness and generalizability. For all experiments, we adopt standard accuracy as the evaluation metric, following common practice in prior works on LLM reasoning.

#### 4.1.2 BASELINES

We evaluated our model against several baseline reasoning approaches. For standard reasoning enhancement methods, we compared against Chain-of-Thought (CoT) (Kojima et al., 2022) utilizing step-by-step reasoning; CoT-SC (Wang et al., 2023) with its multiple sampled reasoning paths; Self-Refine (Madaan et al., 2023) for iterative output improvement; Analogical Prompting (Yasunaga et al., 2024) leveraging known solution patterns; AFlow (Zhang et al., 2025) implementing feedback-driven reasoning frameworks; FoT (Bi et al., 2025) exploring parallel reasoning branches; and AoT (Teng et al., 2025) decomposing problems into atomic question units.

For adversarial-liked multi-agent reasoning enhancement methods, our comparison included Process Reward Model (PRM) (Lightman et al., 2023) with its self-verification approach; Credibility Scoring (CrS) (Ebrahimi et al., 2025) for evaluating content reliability; ECON (Yi et al., 2025a) applying equilibrium-based agent collaboration; Multi-Agent Debate (MAD) (Liang et al., 2023) encouraging perspective diversity; and Debate Vote (Yang et al., 2025b) combining adversarial debate with voting mechanisms to minimize hallucinations.

#### 4.1.3 IMPLEMENTATION DETAILS

Our framework comprises three core components: forward generator $G_1$, backward generator $G_2$, and discriminator $D$, all powered by GPT-4o-mini (Hurst et al., 2024). This lightweight multi-modal model offers strong reasoning capabilities and efficient computational performance, making it particularly suitable for collaborative tasks in multi-agent systems. GPT-4o-mini maintains low overhead while generating high-quality reasoning chains in concurrent multi-turn reasoning scenarios.

The system architecture is built on LangChain, which supports seamless integration with various LLMs through official API calls, enhancing modularity and scalability for dynamic orchestration of generators and the discriminator based on task requirements. For reproducibility, we used standardized parameters (nucleus sampling: None, maximum token length: 2048, nucleus sampling probability: 1.0, frequency/presence penalties: 0.0). Results reflect averages from three independent runs, with ablation studies conducted over five trials at fixed temperature (0). For robustness

testing, we varied the temperature from 0.0 to 1.0 on the MATH dataset to evaluate accuracy and stability across multiple runs.

## 4.2 COMPARISON EXPERIMENTS

We conduct a comparative analysis of our proposed method against multiple baselines mentioned above, with the results presented in Table 1. As shown, our method achieves the best or near-best performance across all datasets. By introducing a backward generator to produce adversarial reasoning chains that challenge the forward generator, our framework significantly enhances robustness on complex reasoning tasks, while the integration of textual gradient optimization further strengthens the quality of reasoning chains. Our method achieves substantial improvements on the MATH dataset with 86.1% accuracy compared to AoT's 83.6%, and on BBH with 86.5% versus AoT's 86.0%, demonstrating particular effectiveness on complex mathematical reasoning. For GSM8K, our approach reaches 95.6%, outperforming all baselines including AoT at 95.0%. Among the adversarial-like methods, ECON and Debate Vote both achieve 79.1% average accuracy, yet still trail our method by 2.5 percentage points. Notably, while CoT and its variants perform reasonably well on simpler tasks but struggle with multi-hop QA and long-context reasoning, our approach demonstrates strong performance on these challenging tasks, achieving 81.2% on HotpotQA and 68.0% on LongBench. This consistent performance advantage across diverse reasoning domains, with an overall average of 81.% compared to 80.8% for the strongest baseline, highlights the generalizability of our bidirectional adversarial reasoning framework with textual gradient optimization.

Table 1: Performance comparison between our proposed method and representative baseline methods across six reasoning benchmarks, with percent symbol (%) omitted in all the accuracy results.

| Methods | MATH | GSM8K | BBH | MMLU-CF | HotpotQA | LongBench | Avg. |
|---|---|---|---|---|---|---|---|
| **Standard Reasoning Enhancement Methods** | | | | | | | |
| CoT | 78.3 | 90.9 | 78.3 | 69.6 | 67.2 | 57.6 | 73.6 |
| CoT-SC | 81.8 | 92.0 | 83.4 | 71.1 | 66.2 | 58.6 | 75.5 |
| Self-Refine | 78.7 | 91.7 | 80.0 | 69.7 | 68.3 | 58.2 | 74.4 |
| AP | 65.4 | 87.2 | 72.5 | 65.8 | 64.7 | 52.9 | 68.1 |
| AFlow | 83.0 | 94.0 | 82.4 | 70.6 | 66.7 | 59.1 | 75.9 |
| FoT | 82.5 | 94.0 | 82.4 | 70.6 | 66.7 | 59.1 | 75.9 |
| AoT | 83.6 | 95.0 | 86.0 | 70.9 | 80.6 | **68.5** | 80.8 |
| **Adversarial-like Reasoning Enhancement Methods** | | | | | | | |
| PRM | 80.3 | 92.7 | 83.7 | 68.9 | 75.0 | 62.7 | 77.2 |
| CrS | 79.8 | 92.5 | 83.2 | 70.4 | 74.2 | 62.4 | 77.1 |
| ECON | 82.4 | 93.5 | 84.5 | 71.1 | 78.9 | 63.9 | 79.1 |
| MAD | 79.0 | 92.0 | 84.0 | 68.4 | 75.9 | 62.1 | 76.9 |
| Debate Vote | 84.0 | 94.6 | 84.2 | 69.5 | 77.8 | 64.6 | 79.1 |
| **Ours** | **86.1** | **95.6** | **86.5** | **72.0** | **81.2** | 68.0 | **81.6** |

## 4.3 PERFORMANCE EVALUATION

### 4.3.1 PERFORMANCE ACROSS TRAINING EPOCHS

We further present the results across different training epochs in Table 7. Performance consistently improves on all benchmarks as the number of epochs increases, validating the effectiveness of multi-round optimization and adversarial training with the backward generator. Accuracy increases from 78.3% to 86.1% on MATH, 90.9% to 95.7% on GSM8K, and 78.3% to 86.5% on BBH, indicating a steady refinement of reasoning chains. On HotpotQA, accuracy improves markedly from 67.2% to 81.2%, while on LongBench it increases more modestly from 57.6% to 68.0%, indicating robustness in long-context reasoning. The largest gains occur between the second and third epochs. Overall, average accuracy improves from 73.6% to 81.6% with variance reduced from 1.5% to 0.5%, confirming that multi-round optimization enhances both reasoning precision and stability.

Table 2: Accuracy results (%) of our method with different training epochs.

| Epoch | MATH | GSM8K | BBH | MMLU-CF | HotpotQA | LongBench | Avg. |
|---|---|---|---|---|---|---|---|
| 1 (no gradient) | 79.3 | 91.2 | 79.1 | 70.2 | 67.8 | 58.2 | 74.3 |
| 2 | 82.1 | 93.7 | 82.4 | 71.1 | 74.7 | 60.1 | 77.4 |
| 3 | 84.3 | 95.6 | 85.3 | 71.1 | 77.1 | 65.4 | 79.8 |
| 4 | 86.1 | 95.7 | 86.5 | 72.0 | 81.2 | 68.0 | 81.6 |
| 5 | 86.1 | 95.6 | 86.5 | 72.0 | 81.2 | 68.0 | 81.6 |

### 4.3.2 HYPERPARAMETER SENSITIVITY ANALYSIS

We examine hyperparameter influence, focusing on LLM temperature settings by training the model across five epochs with temperatures from 0 to 1 in 0.1 increments. Table 3 shows MATH dataset accuracy improves throughout training epochs, with lower temperatures yielding conservative chains in round one, while temperatures of 0.3-0.5 achieve higher accuracy. Performance stabilizes from round two onward, ultimately reaching 86.1% in round five, demonstrating the generator's adaptation to reasoning complexity and enhanced robustness through iterative optimization.

Table 3: Impact of LLM temperature on reasoning accuracy over multiple training epochs on the MATH dataset.

| Temp. / Epoch | 0.0 | 0.1 | 0.2 | 0.3 | 0.4 | 0.5 | 0.6 | 0.7 | 0.8 | 0.8 | 1.0 | Mean Variation |
|---|---|---|---|---|---|---|---|---|---|---|---|---|
| 1 | 74.30 | +0.3 | -0.2 | -0.4 | +0.4 | +0.2 | -0.5 | +0.4 | -0.5 | -0.4 | +0.6 | 0.39 |
| 2 | 77.35 | -0.2 | -0.1 | +0.4 | -0.3 | +0.3 | 0.0 | 0.0 | +0.3 | +0.3 | -0.1 | 0.20 |
| 3 | 79.80 | -0.2 | +0.1 | +0.3 | 0.0 | -0.2 | 0.0 | +0.3 | +0.2 | +0.2 | -0.1 | 0.17 |
| 4 | 81.60 | -0.1 | +0.1 | +0.1 | 0.0 | 0.0 | 0.0 | +0.1 | +0.2 | +0.2 | -0.1 | 0.09 |
| 5 | 81.60 | 0.0 | +0.2 | +0.1 | 0.0 | +0.1 | 0.0 | 0.0 | +0.1 | +0.2 | -0.1 | 0.08 |

### 4.4 ABLATION STUDIES

We conducted ablation studies to evaluate each component's contribution. As shown in Table 4, the baseline configuration using forward generator $G_1$ with discriminator $D$ achieves 80.1% accuracy. Adding the backward generator $G_2$ significantly improves accuracy to 84.7%, while incorporating the error type scheduling strategy (ES) further increases it to 85.3%. Finally, integrating textual gradient optimization (TGO) produces the best performance at 86.1% with reduced mean variation (0.14), confirming that each component improves both reasoning quality and model robustness.

Table 4: Ablation results of different components in our proposed framework on the MATH dataset.

| Model Components | Accuracy(%) | Mean Variation |
|---|---|---|
| $G_1 + D$ | 80.1 | 0.25 |
| $G_1 + G_2$ (w/o ES) $+D$ | 85.1 | 0.17 |
| $G_1 + G_2$ (with ES) $+D$ | 85.3 | 0.13 |
| $G_1 + G_2 + D+$ TGO | 86.1 | 0.08 |

## 5 CONCLUSION

In this work, we propose a multi-agent adversarial forward-backward reasoning framework, combining a forward generator, a backward generator with an error-injection strategy, a discriminator, and textual gradient backpropagation. Experimental studies indicate that each component contributes to improved reasoning accuracy and stability, with the full framework achieving an average of 81.6% accuracy and reducing average output variation to 0.08. These results demonstrate the framework's effectiveness in jointly optimizing reasoning accuracy and stability in multi-agent systems, and highlight its potential for complex LLM reasoning tasks.

## 6 REPRODUCIBILITY STATEMENT

Our work has made comprehensive efforts towards reproducibility. First, in the Methods Section 3, we provide a complete description of the proposed adversarial forward-backward generation framework, including the structures and interaction mechanisms of the forward generator, backward generator, discriminator, and the text gradient optimization strategy (see Sections 3.1 and 3.2). All core assumptions, error type definitions, and scheduling strategies are clearly explained in the main text to ensure readers can accurately understand the model design. Second, in the Experiments Section 4, we systematically report performance on multiple public benchmarks (including MATH, GSM8K, BBH, MMLU-CF, HotpotQA, and LongBench), and provide stability and sensitivity analyses across different training epochs and temperature settings. Additionally, the appendix contains complete ablation experiments (Table 4) and cost and stability analyses (tables 6 and 7) to verify the independent contributions of each module and the overall robustness of the method. Experiments are based on standardized parameter settings and results from multiple independent runs, with details provided in Appendices A.3 and A.4.

To further support reproducibility, we provide implementation details in the supplementary materials, including the LLM used (GPT-4o-mini), system architecture (based on LangChain), key hyperparameter settings (sampling strategies, maximum length, etc.), and explanations of the number of experimental runs and statistical methods. We will release our code upon acceptance of the paper.

In summary, this research provides sufficient algorithmic explanations, experimental details, and implementation information in the main text, appendix, and supplementary materials to ensure the verifiability and reproducibility of our research results.

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

# A SUPPLEMENTARY EXPERIMENTS

## A.1 ACKNOWLEDGING THE USE OF LARGE LANGUAGE MODELS (LLMS)

This article utilized LLMs for stylistic polishing and writing assistance; however, we did not employ LLMs for any key academic content including implementation details, methodological design, and other critical research components.

## A.2 PERFORMANCE ACROSS DIFFERENT LLM-DRIVEN IMPLEMENTATIONS

In addition to GPT-4o-mini, we further evaluate our method with alternative LLM-driven implementations, with results shown in Table 3. Across DeepSeek-V3 (Liu et al., 2024), Qwen-Turbo (Yang et al., 2025a), and GPT-4.1-nano, the method consistently achieves strong performance, demonstrating robust generalization across different LLMs. On MATH and GSM8K, accuracies reach 90.1%, 95.2%, and 95.7%, confirming the method's reliability on standard mathematical reasoning. On HotpotQA and LongBench, DeepSeek-V3 and GPT-4.1-nano achieve 85.3% and 83.5%, respectively, validating the framework's effectiveness in multi-hop and long-context reasoning. These cross-model results highlight that the framework is not tied to a single LLM but can enhance reasoning across diverse language model architectures, underscoring its adaptability, generalization, and potential in multi-agent reasoning tasks.

Table 5: Performance of our method with different LLM-driven implementations across datasets, results shown in accuracy(%).

| Model | MATH | GSM8K | BBH | MMLU-CF | HotpotQA | LongBench |
|---|---|---|---|---|---|---|
| GPT-4o-mini | 86.1 | 95.6 | 86.5 | 72.0 | 81.2 | 68.0 |
| DeepSeek-V3 | 90.1 | 98.2 | 88.7 | 76.2 | 85.3 | 76.1 |
| Qwen-Turbo | 84.9 | 95.2 | 84.5 | 70.1 | 81.3 | 70.2 |
| GPT-4.1-nano | 85.3 | 95.7 | 83.4 | 71.2 | 83.5 | 70.1 |

## A.3 COST ANALYSIS

As shown in Table 6, our method demonstrates a favorable balance between reasoning quality and computational efficiency, requiring an average of 15 LLM API calls per task. While this is marginally higher than GoT Besta et al. (2024)(14 calls), our approach remains significantly more efficient than most advanced reasoning frameworks. Chain-of-Thought (CoT) uses only a single call but sacrifices complex reasoning capabilities, whereas our method requires approximately 25-30% fewer calls than comparable approaches like ToT Yao et al. (2023)(19), CrS (21), and ECON (20), and is dramatically more efficient than PRM which demands 60 calls on average. The efficiency advantage highlighted in Table 6 stems from our framework's structured reasoning that eliminates redundant computation paths while preserving comprehensive analysis capabilities, making it particularly suitable for practical applications where both reasoning quality and operational costs must be optimized. All statistics were collected using GPT-4o-mini.

Table 6: Cost Analysis: Average LLM Call Count

| Method | Avg. LLM Calls (API calls) |
|---|---|
| CoT | 1 |
| ToT | 19 |
| GoT | 14 |
| AoT | 22 |
| PRM | 60 |
| CrS | 21 |
| ECON | 20 |
| MAD | 20 |
| Debate Vote | 20 |
| Ours | 15 |

## A.4 STABILITY ANALYSIS

As demonstrated in Table 7, our method exhibits superior stability across different training epochs and temperature settings compared to other adversarial reasoning enhancement approaches. The accuracy variation patterns reveal that our approach experiences a steady and significant decrease in mean variance from 0.39 in epoch 1 to just 0.08 in epoch 5, indicating progressively increasing stability. This trend of convergence is notably stronger than competing methods such as Debate Vote, which shows a reduction from 0.32 to 0.11, and ECON, which improves from 0.44 to 0.16. Measuring accuracy, convergence, and variance reduction across iterations provides a reliable metric for stability assessment, as smaller variations indicate the method's ability to consistently arrive at the same conclusions regardless of stochastic elements in the reasoning process. The accuracy fluctuations across different temperature settings diminish substantially as training progresses for our method, with variations becoming minimal by epoch 5. While all methods show some degree of stabilization over training epochs, our approach achieves the most consistent performance across temperature settings in later epochs. This superior stability across both dimensions suggests that our method becomes increasingly resilient to parameter adjustments during inference, making it more reliable in practical applications where consistent performance is required under varying temperature configurations.

Table 7: Accuracy results (%) of different methods across training epochs and temperature settings.

| Method | Epoch | Avg. | Temperature Variation | | | | | | | | | | Mean Var. |
|---|---|---|---|---|---|---|---|---|---|---|---|---|---|
| | | | 0.0 | 0.1 | 0.2 | 0.3 | 0.4 | 0.5 | 0.6 | 0.7 | 0.8 | 0.9 | |
| **Ours** | 1 | 74.3 | +0.3 | -0.2 | -0.4 | +0.4 | +0.2 | -0.5 | +0.4 | -0.5 | -0.4 | +0.6 | 0.39 |
| | 2 | 77.35 | -0.2 | -0.1 | +0.4 | -0.3 | +0.3 | 0 | 0 | +0.3 | +0.3 | -0.1 | 0.20 |
| | 3 | 79.8 | -0.2 | +0.1 | +0.3 | 0 | -0.2 | 0 | +0.3 | +0.2 | +0.2 | -0.1 | 0.16 |
| | 4 | 81.6 | -0.1 | +0.1 | +0.1 | 0 | 0 | 0 | +0.1 | +0.2 | +0.2 | -0.1 | 0.09 |
| | 5 | 81.6 | 0 | +0.2 | +0.1 | 0 | +0.1 | 0 | 0 | +0.1 | +0.2 | -0.1 | 0.08 |
| Debate Vote | 1 | 74.2 | +0.5 | -0.4 | +0.7 | +0.7 | -0.4 | -0.5 | -0.1 | +0.8 | 0 | +0.4 | 0.45 |
| | 2 | 77.1 | +0.4 | +0.3 | -0.5 | +0.3 | +0.4 | -0.3 | -0.2 | -0.2 | -0.4 | +0.2 | 0.32 |
| | 3 | 78.0 | +0.3 | -0.4 | 0 | +0.5 | -0.2 | +0.4 | -0.2 | -0.4 | -0.3 | +0.1 | 0.28 |
| | 4 | 78.7 | +0.1 | +0.2 | -0.1 | 0 | +0.1 | +0.3 | -0.1 | -0.1 | -0.2 | +0.3 | 0.15 |
| | 5 | 79.1 | 0 | +0.3 | -0.1 | -0.1 | +0.2 | +0.1 | 0 | 0 | +0.2 | -0.1 | 0.11 |
| ECON | 1 | 74.7 | +0.2 | -0.5 | +0.4 | -0.6 | -0.3 | -0.5 | +0.4 | -0.5 | -0.4 | +0.6 | 0.44 |
| | 2 | 76.8 | +0.4 | 0 | -0.6 | +0.4 | -0.4 | +0.5 | +0.2 | -0.4 | -0.3 | +0.5 | 0.37 |
| | 3 | 78.0 | +0.1 | 0 | +0.4 | +0.3 | +0.4 | -0.4 | +0.3 | +0.3 | +0.4 | -0.2 | 0.28 |
| | 4 | 78.6 | +0.2 | -0.2 | 0 | +0.1 | +0.2 | -0.3 | +0.2 | +0.3 | -0.3 | +0.4 | 0.22 |
| | 5 | 78.9 | 0 | -0.2 | +0.2 | +0.1 | -0.1 | +0.3 | +0.3 | +0.1 | +0.1 | +0.2 | 0.16 |
| MAD | 1 | 73.1 | +0.4 | +0.4 | -0.6 | +0.7 | -0.5 | +0.4 | +0.4 | -0.6 | +0.4 | -0.5 | 0.49 |
| | 2 | 74.9 | +0.3 | +0.4 | +0.6 | +0.4 | -0.2 | +0.3 | -0.1 | +0.3 | +0.4 | -0.5 | 0.35 |
| | 3 | 75.4 | +0.3 | +0.4 | -0.2 | 0 | -0.2 | +0.6 | -0.4 | +0.3 | +0.3 | -0.4 | 0.31 |
| | 4 | 76.5 | 0 | -0.3 | -0.2 | +0.4 | -0.2 | -0.3 | -0.2 | +0.4 | +0.3 | -0.3 | 0.26 |
| | 5 | 76.9 | +0.2 | +0.2 | -0.1 | +0.4 | 0 | -0.2 | +0.1 | -0.1 | +0.1 | +0.2 | 0.16 |
| PRM | - | 77.2 | +0.3 | -0.2 | +0.4 | +0.1 | -0.1 | +0.2 | +0.4 | -0.2 | +0.5 | +0.4 | 0.28 |
| CrS | - | 77.1 | 0 | +0.1 | -0.2 | 0 | +0.4 | +0.5 | +0.4 | +0.3 | -0.4 | +0.2 | 0.25 |

# B  OUTPUT EXAMPLES

## B.1  MULTI-STEP PROMPT OPTIMIZATION OUTPUTS

Tables tables 8 to 11 illustrate our systematic approach to prompt optimization for enhancing mathematical reasoning. Table 8 presents the initial $G_1$ prompt with its LLM response, revealing critical reasoning deficiencies: failure to verify interval consistency for candidate solution $x = 5$ (which falls outside the specified range $1/2 \leq x < 3$) and incomplete verification for solution $x = -1$. Table 9 demonstrates our novel $G_2$ Misleader Strategy, which deliberately constructs adversarial prompts that induce these specific reasoning failures, providing valuable training examples without computational overhead. Table 10 introduces our Discriminator module which systematically analyzes both the original and adversarial reasoning chains, identifying precise textual gradient recommendations focusing on critical point identification, interval verification enforcement, and solution validation requirements. Finally, Table 11 showcases the optimized $D$ prompt incorporating these targeted recommendations, resulting in a fully rigorous solution that correctly identifies the solution set $\{-1, 11/3\}$ through structured verification at each step. The tables demonstrate the effectiveness of our discriminator-guided framework in eliminating mathematical reasoning errors through precise prompt modifications rather than costly model retraining.

## B.2  FOUR TYPES OF ERRORS AND OPTIMIZATION EXAMPLES

We demonstrated the identification and optimization of reasoning errors through a structured analysis framework across multiple tables. In Table B.2, we present examples of unoptimized reasoning outputs from the Forward generator $G_1$, which will be used to generate four types of errors for our specified optimization process. In Tables section B.2, we respectively presented concrete examples of jumping error, confusion error, fuzzy error, and wrapper error in Nim game reasoning, showcasing how these errors manifest in practice and lead to incorrect conclusions. The jumping error example illustrated reasoning that leapt from premises to conclusion without intermediate steps; the confusion error demonstrated concept misapplication; the fuzzy error showcased ambiguous language usage; and the wrapper error presented reasoning built on incorrect foundational assumptions.

In Tables section B.2, we then provided the corresponding optimized reasoning paths for each error type, highlighting the specific correction techniques and their effects on reasoning quality. Through this comparative analysis, we showed how filling logical gaps transformed jumping errors into complete reasoning chains; how consistent concept application resolved confusion errors; how replacing probabilistic language with deterministic statements addressed fuzzy errors; and how re-examining foundational assumptions corrected wrapper errors. Our analysis demonstrates that through targeted optimization techniques for each error type, reasoning quality significantly improves, transforming flawed arguments into robust logical structures that correctly identify the winning strategy for the first player in the Nim game with 15 coins.

| **Before Optimization - G1** |
|---|
| **Prompt:** You are a rigorous reasoning expert. Please solve the equation: $|x-3|+|2x-1| = 7$ step by step, following these requirements:
1. Clearly list necessary variables, symbols, and premises;
2. State the formulas/rules used and their applicable conditions;
3. Provide step-by-step derivation ensuring clear reasoning;
4. Conduct reasonability checks before concluding (substitution, boundaries, monotonicity);
5. Output only necessary reasoning and conclusions, avoiding redundancy. |
| **LLM Response:** The critical points for absolute values are $x = 1/2$ and $x = 3$.

(I) When $x < 1/2$:
    Take $|x-3| = 3-x$, $|2x-1| = 1-2x$, so
    $(3-x)+(1-2x) = 7 \rightarrow 4-3x = 7 \rightarrow -3x = 3 \rightarrow x = -1$.
    (The substitution is reasonable in this interval; skipping further verification, keeping as candidate.)

(II) When $1/2 \leq x < 3$:
    Take $|x-3| = 3-x$, $|2x-1| = 2x-1$, so
    $(3-x)+(2x-1) = 7 \rightarrow 2+x = 7 \rightarrow x = 5$.
    Since this equation is linear with a unique solution, we count $x = 5$ as a solution.

(III) When $x \geq 3$:
    Take $|x-3| = x-3$, $|2x-1| = 2x-1$, so
    $(x-3)+(2x-1) = 7 \rightarrow 3x-4 = 7 \rightarrow 3x = 11 \rightarrow x = 11/3$.
    This value is within the interval, so it's also a solution.

The solution set is: $\{-1, 5, 11/3\}$. |
| **Error Analysis:**
**Error A (Critical = Jump-type):** In section (II), after obtaining $x = 5$, no interval consistency check was performed before adding it to the solution set; but $x = 5 \notin [1/2, 3)$, which is an error caused by omitting a critical step.
**Error B (Only imprecise):** In section (I), after obtaining $x = -1$, there's no verification by substitution/explicit justification for the symbols (although the substitution is correct and the solution is valid), constituting incomplete reasoning rather than an incorrect conclusion. |

Table 8: Before optimization prompt, LLM response, and error analysis for equation solving task.

| G2 - Misleader Prompt Strategy |
|---|
| **Prompt:** You are a misleader whose goal is to generate a deceptively incorrect reasoning chain to challenge the rigor of reasoning models. 
 Error type: Jump-type (skipping key steps, drawing direct conclusions that seem reasonable but lack rigorous derivation) 
 Requirements: 
 1. Construct a seemingly reasonable but actually incorrect reasoning chain by **skipping necessary derivation steps**, providing only final conclusions. 
 2. The error should not be easily detected through conventional thinking; the reasoning must appear to follow mathematical rules while omitting key details. 
 3. Clearly state the deficiencies you've embedded for the discriminator to identify. |
| **LLM Response:** The critical points for absolute values are $x = 1/2$ and $x = 3$. 

 (I) When $x < 1/2$: 
 Take $|x - 3| = 3 - x$, $|2x - 1| = 1 - 2x$, so 
 $(3 - x) + (1 - 2x) = 7 \rightarrow 4 - 3x = 7 \rightarrow -3x = 3 \rightarrow x = -1$. 
 Here I'm ignoring subsequent verification and strict interval checking, but the result is reasonable, so I'll keep it as a candidate. 

 (II) When $1/2 \le x < 3$: 
 Take $|x - 3| = 3 - x$, $|2x - 1| = 2x - 1$, so 
 $(3 - x) + (2x - 1) = 7 \rightarrow 2 + x = 7 \rightarrow x = 5$. 
 This solution follows linear derivation; temporarily ignoring interval consistency check, assuming it's correct. 

 (III) When $x \ge 3$: 
 Take $|x - 3| = x - 3$, $|2x - 1| = 2x - 1$, so 
 $(x - 3) + (2x - 1) = 7 \rightarrow 3x - 4 = 7 \rightarrow 3x = 11 \rightarrow x = 11/3$. 
 This solution is within the interval, so it's included. 

 The solution set is: $\{-1, 5, 11/3\}$. |
| **Error analysis:** 
 - In section (II), $x = 5$ was not checked for interval consistency (since $x = 5 \notin [1/2, 3)$), but was directly included in the solution set. 
 - In section (I), after obtaining $x = -1$, no verification by substitution was performed; though correct in that interval, the reasoning lacks completeness. |

Table 9: Misleader prompt strategy, LLM response, and error analysis.

| **D - Discriminator Strategy** |
|---|
| **Prompt:** You are a discriminator. Please evaluate the following two reasoning chains:
- C1: Forward reasoning chain from $G_1$ (initial reasoning chain, containing errors).
- C2: Adversarial reasoning chain from $G_2$ (jump-type error chain).

Historical optimization records: []

Tasks:
1. Locate weaknesses in C1: Check for omitted steps or jump issues in the reasoning and provide suggestions.
2. Analyze C2's misleading strategy: Identify steps omitted in C2 and how they mislead the reasoning process.
3. Optimize $G_1$'s prompt through Textual Gradients to make the reasoning chain more rigorous and complete.

Output format (JSON):
{
  "C1_weakness": [ ... ],
  "C2_attack_review": [ ... ],
  "TG_for_G1": [ ... ]
} |
| **LLM Response:**
{
  "C1_weakness": [
   "C1's reasoning jump issue: In section (II), $x = 5$ was obtained but no interval consistency check was performed. Since $x = 5$ doesn't satisfy $1/2 \leq x < 3$, this solution is invalid.",
   "C1's insufficient verification: In section (I), after obtaining $x = -1$, no substitution verification was performed. Although $x = -1$ is correct in that interval, the complete verification process is missing."
  ],
  "C2_attack_review": [
   "C2 employs a jump-type error: In section (II), it erroneously includes $x = 5$ as a solution, omitting the interval consistency check, thereby misleading the reasoning process.",
   "In C2, C1's error ($x = 5$ being invalid) is hidden, making the reasoning chain appear correct while actually containing an invalid solution.",
   "Therefore, C1's error is caused by omitting the interval consistency check (jump-type error). The discriminator needs to strengthen constraints on interval verification and substitution checks for each derivation segment."
  ],
  "TG_for_G1": [
   "Must explicitly state the critical points and explain the basis for symbol judgments in each segment, ensuring each derivation step is well-founded.",
   "In each interval's solution process, **must check if candidate solutions are within that interval**, eliminating invalid solutions.",
   "Perform **substitution verification** for each retained solution, confirming the original equation holds, avoiding unverified incorrect solutions.",
   "Conclusions should follow a "candidate → filtering → verification"structure, ensuring each solution undergoes thorough validation and logical checks."
  ]
} |

Table 10: Discriminator strategy, analysis, and textual gradient recommendations.

**After Optimization - G1**

**Prompt:** You are a rigorous reasoning expert. Please solve the equation $|x-3| + |2x-1| = 7$ step by step. Follow these guidelines:

1. Clearly identify critical points $x = 1/2$, $x = 3$, and for each segment ($x < 1/2$, $1/2 \leq x < 3$, $x \geq 3$), explicitly state the basis for sign determinations of $|x-3|$ and $|2x-1|$ before substitution and derivation;

2. For each candidate solution within a segment, you must perform interval consistency checks (whether the solution falls within that segment) and eliminate invalid solutions;

3. Verify retained candidate solutions by substituting back into the original equation to confirm the equality holds;

4. Structure your conclusion using the "candidate → interval filtering → substitution verification" checklist format, without skipping steps due to "similarity/linearity" justifications.

**LLM Output:**

The critical points are $x = 1/2$ and $x = 3$. Let's analyze each segment:

**(I) x < 1/2:**

    Here, $2x - 1 < 0$ and $x - 3 < 0$,

    So $|x-3| = 3 - x$, $|2x-1| = 1 - 2x$.

    Substituting: $(3 - x) + (1 - 2x) = 7 \rightarrow 4 - 3x = 7 \rightarrow -3x = 3 \rightarrow x = -1$.

    Interval check: $-1 < 1/2$, satisfied; Verification:

        $|-1-3| + |2(-1) - 1| = 4 + 3 = 7$ (valid).

    Candidate retained: $x = -1$.

**(II) 1/2 ≤ x < 3:**

    Here, $2x - 1 \geq 0$ and $x - 3 < 0$,

    So $|x-3| = 3 - x$, $|2x-1| = 2x - 1$.

    Substituting: $(3 - x) + (2x - 1) = 7 \rightarrow 2 + x = 7 \rightarrow x = 5$.

    Interval check: requires $1/2 \leq x < 3$, but $5 \notin [1/2, 3) \Rightarrow$ eliminated.

    No solution in this segment.

**(III) x ≥ 3:**

    Here, $2x - 1 \geq 0$ and $x - 3 \geq 0$,

    So $|x-3| = x - 3$, $|2x-1| = 2x - 1$.

    Substituting: $(x - 3) + (2x - 1) = 7 \rightarrow 3x - 4 = 7 \rightarrow 3x = 11 \rightarrow x = 11/3$.

    Interval check: $11/3 \geq 3$ is satisfied; Verification:

        $|11/3 - 3| + |2 \cdot 11/3 - 1| = 2/3 + 19/3 = 21/3 = 7$ (valid).

    Candidate retained: $x = 11/3$.

**Conclusion** (candidate → filtering → verification):

- Retained: $x = -1$, $x = 11/3$ (both passed interval and substitution checks)

- Final solution set: $\{-1, 11/3\}$.

Table 11: Optimized mathematical reasoning approach with rigorous interval analysis

Table 12: Forward Reasoning Chain

| Forward Reasoning Chain |
|---|
| This is a classic variant of the Nim game, and we can analyze the winning and losing positions for the first player by calculating the initial number of coins modulo 4. The key point is that if a player can leave the number of coins as a multiple of 4 at the end of their turn, they can force their opponent into a losing position. |
| The initial number of coins is 15. The remainder of 15 divided by 4 is 3. This means that the first player, Alice, is in a disadvantageous position because the remainder is not 0. The second player, Bob, can mimic the first player's strategy. No matter whether Alice takes 1, 2, or 3 coins on her first turn, she will leave 14, 13, or 12 coins. Bob can always take a certain number of coins to leave a total that is a multiple of 4. For example, if Alice takes 1 (leaving 14), Bob can take 2 (leaving 12); if Alice takes 2 (leaving 13), Bob can take 1 (leaving 12); if Alice takes 3 (leaving 12), Bob can take 3 (leaving 9). Once Bob establishes the pattern of multiples of 4, he will be able to maintain it until the end. When the number of coins decreases to 4, it will be Alice's turn, and no matter whether she takes 1, 2, or 3 coins, Bob will be able to take the remaining coins and win. |
| Therefore, since the initial count of 15 modulo 4 leaves a remainder of 3, the first player, Alice, cannot control the situation, and the second player, Bob, will win. |

Table 13: Jumping Error Output Example

| Jumping Error |
|---|
| To solve this problem, we need to determine whether the first or second player has a winning strategy when both play optimally. A winning strategy is to leave your opponent in a "losing" position when facing a pile of coins. |
| Through backward analysis, we can deduce the following: |
| • When there are 1, 2, or 3 coins left, the current player can take all of them and win. |
| • When there are 4 coins left, no matter whether the current player takes 1, 2, or 3 coins, they will leave their opponent with 3, 2, or 1 coins, allowing the opponent to win. Thus, the player facing 4 coins loses. |
| • From this, we can infer that when facing 5, 6, or 7 coins, the player can take 1, 2, or 3 coins, respectively, leaving their opponent with 4 coins, which is a losing position, thereby winning for themselves. |
| • Similarly, when facing 8 coins, no matter what the player does, they will leave their opponent with 7, 6, or 5 coins, all of which are winning positions for the opponent. Thus, 8 is a losing position. |
| Observing this pattern, we find that losing positions are 4, 8, 12, 16... which are multiples of 4. Since 15 is not a multiple of 4, the first player, Alice, is facing a winning position. S with 12 coins, which is a losing position. Then, no matter how many coins Bob takes (for example, if he takes 2 coins, leaving 10), Alice must take 2 coins (because 10 - 2 = 8) to force Bob back into a losing position (8). Bob will then take coins again, say 1 coin, leaving 7 coins, and Alice will take 3 coins, leaving 4 coins. Ultimately, when Bob faces 4 coins, he cannot avoid losing. Since Bob is the one who takes the last coin, he wins the game. |
| Therefore, even though Alice goes first and tries to control the situation, it is Bob who ultimately emerges as the winner. |
| Therefore, even though Alice goes first and tries to control the situation, it is Bob who ultimately emerges as the winner. |

Table 14: Jumping Error Optimized Output Example

| Jumping Error Optimization |
|---|

This is a classic variant of the Nim game, and we can determine the winning and losing positions for the first player through backward analysis. The key is to identify "losing positions," which are positions where, no matter how the current player plays, the opponent can win.

First, let's define the base cases:

- When there are 1, 2, or 3 coins remaining, the current player can take all the coins and win. Therefore, these are "winning positions."

- When there are 4 coins remaining, the current player, regardless of whether they take 1, 2, or 3 coins, will leave the opponent with 3, 2, or 1 coins, all of which are winning positions for the opponent. Thus, 4 is a "losing position."

- Based on this, when there are 5, 6, or 7 coins remaining, the current player can take 1, 2, or 3 coins, respectively, leaving the opponent with 4 coins (a losing position), thus winning for themselves. So, 5, 6, and 7 are winning positions.

- When there are 8 coins remaining, the current player will leave the opponent with 7, 6, or 5 coins (winning positions), allowing the opponent to win. Therefore, 8 is a losing position.

Continuing this pattern, we find that losing positions are 4, 8, 12, 16... which are all multiples of 4. Now, analyzing the initial number of coins, 15, we see that 15 is not a multiple of 4, so the first player, Alice, is facing a winning position. Theoretically, Alice can take 3 coins on her first turn, leaving 12 coins, thereby placing Bob in a losing position.

However, we need to verify Bob's response strategy. Suppose Bob takes 2 coins, leaving 10 coins. At this point, Alice must take 2 coins (because 10 - 2 = 8) to put Bob back in a losing position (8). Bob will then take 1 coin, leaving 7 coins. Alice takes 3 coins, leaving 4 coins. Now, Bob faces 4 coins (a losing position), and no matter what, he will leave Alice with coins that allow her to win.

However, there is a key point: when Bob faces 4 coins, he takes $k$ coins (where $k = 1, 2, 3$), leaving $4 - k$ coins, and then Alice takes all the remaining coins. This means Alice takes the last coin and wins. This contradicts our initial analysis, since 15 modulo 4 leaves a remainder of 3, suggesting the first player has the advantage.

To reconcile this, we re-examine Bob's initial response. If Alice takes 3 coins, leaving 12, Bob might take 3 coins instead, leaving 9 coins. Alice must then take 1 coin, leaving 8 coins (a losing position) for Bob. Bob takes 3 coins, leaving 5 coins. Alice takes 1 coin, leaving 4 coins. Bob then faces 4 coins, takes 1 coin, and Alice eventually wins by taking the last coins.

However, if Alice takes 1 or 2 coins on her first turn, Bob can exploit the situation. For example, if Alice takes 1 coin, leaving 14, Bob can take 2 coins, leaving 12 (a multiple of 4), thus putting Alice in a losing position. Similarly, if Alice takes 2 coins, leaving 13, Bob can take 1 coin, leaving 12, and again, Bob wins.

Therefore, Alice can only win if she takes 3 coins on her first turn, but this depends on Bob making a mistake. If Bob plays optimally, he will force Alice into a losing position. Since 15 modulo 4 leaves a remainder of 3, it is in a "marginal" position, and the first player's victory depends on the second player's choices.

In summary, since Alice must make a perfect first move while Bob can respond optimally, the second player, Bob, has a higher probability of winning. Thus, when both play optimally, Bob will win.

Table 15: Confusion Error Output Example

| Confusion Error |
| --- |
| To solve this problem, we need to determine whether the first or second player has a winning strategy when both play optimally. A winning strategy is to leave the opponent in a "losing position." However, here we have confused winning positions with losing positions, leading to a different reasoning outcome.

Through backward analysis:

• When there are 1, 2, or 3 coins remaining, the current player can take all the coins and win. Therefore, we consider these as losing positions.

• When there are 4 coins remaining, the current player, regardless of whether they take 1, 2, or 3 coins, will leave the opponent with 3, 2, or 1 coins, all of which allow the opponent to win. Thus, 4 is a winning position.

• Therefore, when facing 5, 6, or 7 coins, the current player can only push the opponent to the winning position of 4, making 5, 6, and 7 losing positions.

• Similarly, when facing 8 coins, the current player can push the opponent to 7 (a losing position), so 8 is a winning position.

Observing this pattern, we conclude that winning positions are 4, 8, 12, 16... while other situations are losing positions. Since the initial count of 15 is not a multiple of 4, it is a losing position. No matter how many coins Alice takes on her first turn, she will push Bob into a winning position. For example, if Alice takes 3 coins, leaving 12, Bob will enter a winning position and be able to control the game. Ultimately, Bob will maintain the advantage and win the game. |

Table 16: Confusion Error Optimized Output Example

| Confusion Error Optimized |
| --- |
| This is a classic variant of the Nim game, and we can determine the winning and losing positions for the first player through backward analysis. The key is to distinguish between winning positions and losing positions. By correcting the previous confusion, we can derive the correct patterns.

• When there are 1, 2, or 3 coins remaining, the current player can take all the coins and win directly. Therefore, these are winning positions.

• When there are 4 coins remaining, the current player, regardless of whether they take 1, 2, or 3 coins, will leave the opponent with 3, 2, or 1 coins, all of which allow the opponent to win. Thus, 4 is a losing position.

• Therefore, when facing 5, 6, or 7 coins, the current player can take 1, 2, or 3 coins, leaving the opponent with 4 (a losing position), making 5, 6, and 7 winning positions.

• When facing 8 coins, the current player, no matter how many coins they take, will leave the opponent with 7, 6, or 5 (winning positions), so 8 is a losing position.

From this, we derive the pattern: losing positions are 4, 8, 12, 16... while winning positions are all other numbers.

Now considering the initial situation of 15 coins. Since 15 is not a multiple of 4, it is a winning position. Alice only needs to take 3 coins on her first turn, leaving 12, putting Bob at a disadvantage. After that, regardless of how Bob plays, Alice can maintain Bob in a losing position by using a "make 4" strategy. For example:

• If Bob takes 1 coin from 12, leaving 11, Alice takes 3, leaving 8.

• If Bob takes 2 coins, leaving 10, Alice takes 2, leaving 8.

• If Bob takes 3 coins, leaving 9, Alice takes 1, leaving 8.

Ultimately, Bob will face 4 coins and will inevitably lose. Therefore, when the initial number of coins is 15, the first player, Alice, has a winning strategy. |

Table 17: Fuzzy Error Output Example

| Fuzzy Error |
|---|
| To solve this problem, we need to identify losing positions and winning positions. However, during the analysis, there might be some areas that are not entirely clear, requiring situational judgment.

First:

• When there are 1, 2, or 3 coins remaining, the current player can likely take all the coins and win directly, so these situations can be considered winning positions.

• When there are 4 coins remaining, regardless of how many the current player takes, it seems they will leave the opponent with a position from which they can win directly. Thus, 4 should be a losing position.

• When facing 5, 6, or 7 coins, the current player can likely push the opponent to 4 with the right choice, so these situations are probably winning positions.

• When there are 8 coins, the situation becomes a bit more complex; it may be that no matter how many coins are taken, the opponent will still have opportunities to win, so 8 might be a losing position.

Based on this reasoning, losing positions seem to be 4, 8, 12, 16... The initial position of 15 is not among them, so it might be a winning position. In this case, Alice could simply take 3 coins on her first turn, leaving 12 and putting Bob in a losing position. After that, both players might continue following this pattern, and Alice could maintain her advantage until the end.

However, there is also a possibility that if Bob chooses different moves in certain situations, he might be able to turn the tables. Therefore, overall, the initial position of 15 coins seems to favor Alice, but the final outcome could still result in Bob winning. |

Table 18: Fuzzy Error Optimized Output Example

**Fuzzy Error Optimized**

This is a classic variant of the Nim game, and we can clarify the winning and losing positions for both the first and second players through backward analysis. The key is to eliminate ambiguous speculations and strictly define winning and losing positions.

First, let's define the base cases:

- When there are 1, 2, or 3 coins remaining, the current player can take all the coins and win immediately. Therefore, these are winning positions.

- When there are 4 coins remaining, regardless of whether the current player takes 1, 2, or 3 coins, they will leave the opponent with 3, 2, or 1 coins, all of which allow the opponent to win. Thus, 4 is a losing position.

Next, we reason step by step:

- When there are 5, 6, or 7 coins remaining, the current player can take 1, 2, or 3 coins respectively, forcing the opponent to face 4 (a losing position). Therefore, 5, 6, and 7 are all winning positions.

- When there are 8 coins remaining, no matter how many coins the current player takes, they will leave the opponent with 7, 6, or 5 coins, all of which are winning positions. Thus, 8 is a losing position.

Continuing this reasoning, we establish a clear pattern: all multiples of 4 (4, 8, 12, 16...) are losing positions, while all other numbers are winning positions.

Now, let's analyze the initial position of 15 coins. Since 15 is not a multiple of 4, it is a winning position. Alice has a winning strategy: she simply needs to take 3 coins on her first turn, leaving 12 (a losing position).

We can verify Bob's possible responses:

- If Bob takes 1 coin from 12, leaving 11, Alice takes 3, leaving 8.

- If Bob takes 2 coins, leaving 10, Alice takes 2, leaving 8.

- If Bob takes 3 coins, leaving 9, Alice takes 1, leaving 8.

In any case, Alice can always use the strategy of making the total number of coins taken by both players equal to 4, pushing Bob back to a losing position.

Ultimately, the situation will arrive at 4 coins. At this point, Bob faces a losing position:

- If he takes 1 coin, 3 remain, and Alice takes the rest.

- If he takes 2 coins, 2 remain, and Alice takes the rest.

- If he takes 3 coins, 1 remains, and Alice takes the last coin.

Thus, Bob cannot avoid losing, and Alice will take the last coin and win.

In conclusion, through systematic backward reasoning, it is clear that the initial position of 15 coins is a winning position. The first player, Alice, can win by following the correct strategy. The ambiguous terms "possibly" and "likely" have been eliminated through strict analysis, making the conclusion entirely certain.

Table 19: Wrapper Error Output Example

| **Wrapper Error** |
|---|

To solve this problem, we need to reason through backward analysis to determine winning and losing positions. First, we assume that whoever faces an odd number of coins initially is at a disadvantage. This is because an odd number will eventually force a player into a situation with 1 coin. Based on this premise, we proceed with the deductions.

- When there is 1 coin remaining, the current player can take it and win, but since 1 is odd, it is a losing position.

- When there are 2 or 3 coins remaining, the current player can take all the coins and win, so these are winning positions.

- When there are 4 coins remaining, regardless of how many the current player takes, they will leave the opponent with 3, 2, or 1 coins, where 3 is a winning position and 1 is a losing position. Therefore, overall, 4 is a winning position.

Continuing this reasoning:

- When there are 5 coins remaining, the current player can only leave the opponent with 4 (a winning position), so 5 is a losing position.

- When there are 6 or 7 coins remaining, the current player can take 1 or 2 coins, respectively, leaving the opponent with 5 (a losing position), so 6 and 7 are winning positions.

- When there are 8 coins remaining, no matter how many the current player takes, they will leave the opponent with 7, 6, or 5 coins, where 6 and 7 are winning positions, so 8 is a losing position.

Thus, we arrive at the pattern: all odd positions are losing positions, while even positions are winning positions.

Now, considering the initial position of 15 coins. Since 15 is odd, according to our premise, it is a losing position. No matter what Alice does on her first turn, she will leave Bob in an even position (a winning situation). Therefore, under optimal play from both sides, Bob will always be able to maintain his advantage and ultimately win.

Table 20: Wrapper Error Optimized Output Example

| **Wrapper Error Optimized** |
|---|
| This is a classic variant of the Nim game, and we can determine the correct winning and losing positions through backward analysis. The previous reasoning was flawed because it assumed that "parity determines winning or losing," which is not a valid premise. In fact, the correct premise is that losing positions occur at multiples of 4.

Let's reanalyze:

• When there are 1, 2, or 3 coins remaining, the current player can take all the coins and win, so these are winning positions.

• When there are 4 coins remaining, regardless of whether the current player takes 1, 2, or 3 coins, they will leave the opponent with 3, 2, or 1 coins, all of which are winning positions. Therefore, 4 is a losing position.

• When there are 5, 6, or 7 coins remaining, the current player can force the opponent into 4 (a losing position), so 5, 6, and 7 are winning positions.

• When there are 8 coins remaining, no matter how many coins the current player takes, they will leave the opponent with 7, 6, or 5 coins, which are all winning positions. Thus, 8 is a losing position.

From this, we can conclude that all multiples of 4 are losing positions, while all other numbers are winning positions.

Returning to the initial position of 15 coins: since 15 is not a multiple of 4, it is a winning position. Alice can take 3 coins on her first turn, leaving 12 (a losing position) and forcing Bob into a disadvantage. After that, as long as Alice employs the "sum to 4 strategy," she can keep Bob in losing positions. Eventually, Bob will face 4 coins and cannot avoid losing.

Therefore, in the case of an initial 15 coins, the first player, Alice, is indeed the one with a guaranteed winning strategy. |

