# OpenReview forum: "Enhancing Reasoning Chains through GANs and Textual Gradient Feedback"
_ICLR.cc/2026/Conference — ICLR 2026 Conference Withdrawn Submission_

### Official Review · Reviewer_qDHT · 2025-10-30

**Soundness:** 3
**Presentation:** 3
**Contribution:** 3
**Rating:** 6
**Confidence:** 4

**Summary:**

The paper proposes an adversarial framework to optimize prompts for LLM reasoning. Two generators tackle the same question independently: a forward solver (G1) aims to produce a correct chain of thought from its prompt, while an adversarial solver (G2) generates a plausible-but-wrong chain from its own prompt by injecting targeted error types (e.g., skipped checks, concept confusions). A discriminator (D) reads both chains and returns structured, natural-language feedback (“textual gradients”) that is used to rewrite the prompts of G1, G2, and D themselves. Iterating this per question—and across a dataset—yields an increasingly strong solver prompt that is more robust to common failure modes, without requiring ground-truth solutions.

**Strengths:**

The work is original in introducing adversarial prompt optimization without requiring ground truth reasoning chains. The method is well presented and empirically validated across multiple benchmarks, with clear accuracy improvements.

**Weaknesses:**

1) The paper is unclear about whether prompt optimization is performed per-sample, per-batch, or globally across the dataset. This is a critical implementation detail affecting both computational cost and how broadly the learned prompts can generalize.
2) The use of quasi-GAN terminology is somewhat misleading: the framework does not involve a minimax objective in which a generator attempts to fool a discriminator. Instead, G1 is refined to improve reasoning correctness, while G2 is not trained to defeat or mislead the discriminator, but rather to surface plausible failure modes. The discriminator provides textual feedback that guides both toward more informative contrasts, making the interaction closer to contrastive self-play than GAN-style adversarial training.
3) Baseline comparisons across models are limited, as the baseline methods are only reported for GPT-4o-mini. Table 5 reports the proposed method on additional LLMs but does not include the reference baselines reported in Table 1 (e.g., CoT, Self-Refine, PRM, etc.). As a result, we cannot assess comparative performance or cross-model generality.
4) The three claimed contributions—accuracy, stability, and robustness—are unevenly supported. Accuracy gains are demonstrated across benchmarks (Table 1; Table 2). Stability is evaluated only on MATH and against a subset of adversarial-style baselines via temperature/variance analyses (Tables 3 & 7). By contrast, robustness to prompt perturbations is claimed (e.g., in the Abstract: "increases robustness to prompt perturbation" and the statement that “the core idea is to improve the robustness of G1,” line 113)  but it is not quantitatively measured with dedicated prompt-perturbation tests.
(5) The paper does not sufficiently position itself relative to closely related dual-agent adversarial reasoning frameworks, particularly Agentic Adversarial QA for Improving Domain-Specific LLMs [1]. That work also uses a two-agent setup with TextGrad-based feedback to expose model weaknesses, though with a different adversarial interaction and optimization objective. A clearer comparison would help clarify what is novel about the proposed approach and how it differs conceptually from prior dual-agent self-improvement methods.

Minor typos and inconsistencies:
- In §4.4 text, “Adding the backward generator G2 significantly improves accuracy to 84.7%” should be 85.1% per the ablation table. See Table 4 (“G1+G2 (w/o ES)+D = 85.1”).
- The Abstract says accuracy improves “from 73.7% to 81.6%,” but the paper elsewhere reports 73.6% → 81.6% (and Table 1’s strongest baseline average is 80.8 with our method 81.6). Please reconcile 73.7 vs 73.6.
- In the main text, “with an overall average of 81.% compared to 80.8%” is missing a digit; it should be 81.6% vs 80.8% (as in Table 1).
- Table 3 header duplicates “0.8” in the temperature columns; please remove the duplicate.
- Table 3 caption incorrectly states “on the MATH dataset”; the reported values do not match Table 1’s MATH results. The caption should clarify what data the table reflects; I assume this corresponds to the aggregated benchmark rather than MATH specifically.


[1] Grari, V., Tomoiaga, C., Lamprier, S., Hashimoto, T., & Detyniecki, M. (2025). Agentic Adversarial QA for Improving Domain-Specific LLMs. In Second Workshop on Test-Time Adaptation: Putting Updates to the Test! at ICML 2025. https://openreview.net/forum?id=5tAyBL4nF8

**Questions:**

1) Are the optimized prompts P_1, P_2 and the discriminator prompt shared across all test samples, per-batch or is the optimization performed individually for each sample?

2) Do you use a separate train/test split, or is the method applied directly to the test set to improve the chains for those questions? If it’s the latter, what do “training epochs” in §4.3.1 and Table 2 refer to (e.g., iterations of self-play per test item vs. repeated passes on the test set)?

3) Table 5 reports the proposed method on additional LLMs, but does not include the reference baselines reported in Table 1 (e.g., CoT, Self-Refine, PRM, etc.). Do you also have these baseline results for the other models shown in Table 5?

4) The paper states that the method improves robustness to prompt perturbations (Abstract; main text). However, I did not see a dedicated evaluation of robustness in the experiments. Could you clarify where robustness is demonstrated, or provide more detail on how it was assessed? If this aspect was not evaluated directly, clarifying the scope of the claim would help readers interpret the results appropriately.

---

### Official Review · Reviewer_bgcv · 2025-10-31

**Soundness:** 1
**Presentation:** 1
**Contribution:** 2
**Rating:** 2
**Confidence:** 4

**Summary:**

This paper proposes a novel adversarial framework, termed Quasi-GAN, to improve the reasoning accuracy and stability of Large Language Models (LLMs). The framework consists of a forward generator (G1) that produces correct reasoning chains, a backward generator (G2) that deliberately creates structured, erroneous reasoning chains, and a discriminator (D) that evaluates both. A key contribution is the use of "Textual Gradient Optimization" (TGO), where the discriminator provides natural language feedback (textual gradients) to iteratively refine the prompts of the generators. The authors demonstrate through extensive experiments that their method achieves state-of-the-art results on several reasoning benchmarks, while also significantly reducing output instability.

**Strengths:**

- The core idea of using a backward generator (G2) to create structured, adversarial reasoning chains is novel and compelling. This moves beyond simple self-correction and introduces a dynamic, competitive training environment.
- The ablation study in Table 4 provides strong evidence that the backward generator is the primary driver of the performance gains, validating the central hypothesis of the paper.
- The introduction of an error scheduling strategy (a form of curriculum learning for the backward generator) is a thoughtful and well-reasoned component of the methodology.

**Weaknesses:**

- The technical description of Textual Gradient Optimization (TGO) is a major weakness. Section 3.3 presents the mechanism using formal chain rule notation (Equations 5-7), suggesting a rigorous, gradient-based optimization process. However, based on the description this is merely a conceptual analogy. The actual process appears to be a standard critique-and-revise loop, where an LLM is prompted to rewrite a prompt based on natural language feedback. This presentation is misleading, overstates the technical novelty, and lacks the scientific rigor expected at a top-tier venue.
- The base model used to generate results in tables 1 through 4 is unclear. Also, it seems to be a single base model. To show generalizable performance, it is critical to show results of the approach with multiple models.

**Questions:**

- Can you confirm if a standard critique-and-revise loop was used based on the feedback or actual gradient-based optimization was performed. If yes, does this limit using the framework to open-source models?
- Can you provide results with multiple base models?

---

### Official Review · Reviewer_nXjX · 2025-10-31

**Soundness:** 3
**Presentation:** 1
**Contribution:** 3
**Rating:** 4
**Confidence:** 3

**Summary:**

The paper presents a method for optimizing reasoning chains using a strategy inspired by adversarial training. The setup assumes a multi-agent system in which one LLM acts as a generator, producing a correct answer to a given prompt, while another LLM generates a suboptimal or incorrect answer to the same prompt. A third LLM, serving as a discriminator, compares the two answers and provides feedback on how each can be improved. This feedback is then propagated to the input prompts through TextGrad. The authors conduct experiments using multiple LLMs (mostly closed-source ones) across several benchmarks.

**Strengths:**

1. The core idea of the paper is novel and appealing. Using adversarial training to enhance reasoning quality is an elegant and creative approach.
2. The integration of TextGrad is a strength, as it enables optimization with closed-source models. However, since the main contribution lies in the training strategy rather than model access, it would have been useful to include comparisons with open models or LoRA-based fine-tuning (see weaknesses).

**Weaknesses:**

1. While the use of TextGrad for optimizing GPT prompts is clever, the paper’s primary claims concern the general training method. From my understanding of TextGrad, it should be possible to optimize network weights without altering the training logic. If this is correct, why not pursue such an approach on open models? It would also enable a fairer comparison with baseline methods.
2. Since the method relies on TextGrad backpropagation, I think it would make sense to include a baseline using a STAR-style [1] approach with the same network. This comparison should also be feasible in the GPT-based setup proposed by the authors since GPT supports finetuning. More broadly, maybe the paper should include comparisons with trained baselines such as MALT on open networks, which also use non–training-free setups.
3. The cost analysis focuses only on API calls. However, the optimization process also requires computation time, raising questions about the fairness of the comparison. This concern is especially relevant because the performance gains over methods such as AoT are relatively modest. Therefore, it is important to report and compare the end-to-end processing times from question to answer.
4. The presentation of the manuscript could be improved. The problem formulation is unclear, and framing the setup as multi-agent is somewhat misleading, as this structure is only relevant during training, while inference involves only the generator (G1). Additionally, the related work section is very brief and does not provide a comprehensive overview of the existing literature, which makes it difficult for readers less familiar with LLM research to contextualize the contribution.

[1] STAR: Bootstrapping Reasoning With Reasoning

**Questions:**

I would consider my expertise in this field at a medium level, so the authors should feel free to reply to my questions textually and without experiments if they feel like my criticism is not justified. I will be happy to engage in discussions.

1. Why not having open baselines that would allow us to understand the effectiveness of the proposed adversarial training? In my understanding, that's the major contribution and as such it necessitates of proper comparisons with trained methods.
2. Is it possible to conduct a more comprehensive cost analysis?

---

### Note · Authors · 2025-11-15

I have read and agree with the venue's withdrawal policy on behalf of myself and my co-authors.